# Centrality-Dependent Lévy HBT Analysis in $\sqrt{s_{NN}} = 5.02$ TeV PbPb Collisions at CMS

**Balázs Kórodi on behalf of the CMS Collaboration**

Department of Atomic Physics, Faculty of Science, Eötvös Loránd University, Pázmány Péter Sétány 1/A, H-1111 Budapest, Hungary; balazs.korodi@cern.ch

**Abstract:** The measurement of two-particle Bose–Einstein momentum correlation functions are presented using $\sqrt{s_{NN}} = 5.02$ TeV PbPb collision data, recorded by the CMS experiment in 2018. The measured correlation functions are discussed in terms of Lévy-type source distributions. The Lévy source parameters are extracted as functions of transverse mass and collision centrality. These source parameters include the correlation strength $\lambda$, the Lévy stability index $\alpha$, and the Lévy scale parameter $R$. The source shape, characterized by $\alpha$, is found to be neither Gaussian nor Cauchy. A hydrodynamic-like scaling of $R$ is also observed.

**Keywords:** heavy ions; quark–gluon plasma; femtoscopy; Lévy HBT





## 1. Introduction

The investigation of the femtometer-scale space–time geometry of high-energy heavy-ion collisions has been an important area, called femtoscopy, of high-energy physics for several decades [1]. The main idea of this field originates from astronomy, since it is analogous with the well-known Hanbury Brown and Twiss (HBT) effect that describes the intensity correlation of photons [2,3]. In high-energy physics, however, the observable is the quantum-statistical momentum correlation of hadrons, which carries information about the femtometer-scale structure of the particle-emitting source [4,5]. The measurements of such momentum correlations are partially responsible for establishing the fluid nature of the quark–gluon plasma (QGP) created in heavy-ion collisions [6,7]. Furthermore, the measured source radii provide information about the transition from the QGP to the hadronic phase [8,9], as well as about the phase space of quantum chromodynamics [10].

Recent high-precision femtoscopic measurements [11,12] have shown that the previously widely assumed Gaussian [6,13,14] or Cauchy [15,16] source distributions do not provide an adequate description of the measured correlation functions. Instead, a generalization of these distribution, the Lévy alpha-stable distribution [17], is needed for a statistically acceptable description [11,12]. The shape of the Lévy distribution is characterized by the Lévy stability index $\alpha$, and can be influenced by various physical phenomena, e.g., anomalous diffusion [18–20], resonance decays [21,22], jet fragmentation [23], and critical phenomena [24]. Until now, the $\alpha$ parameter had not been measured at the largest energies accessible at the LHC. The question of how $\alpha$ changes compared to lower energies signifies the need for a Lévy HBT analysis at LHC energy.

In this paper, the Lévy HBT analysis of two-particle Bose–Einstein momentum correlations is presented using $\sqrt{s_{NN}} = 5.02$ TeV PbPb collision data recorded by the CMS experiment. The source parameters, extracted from the correlations functions, are studied as functions of transverse mass and collision centrality.

## 2. Femtoscopy with Lévy Sources

The quantum-statistical momentum correlation of identical bosons is called Bose–Einstein correlation. This correlation is in connection with the source function $S(x, p)$ [4,5],

which is the phase-space probability density of particle production at space–time point $x$ and four-momentum $p$. After some approximations detailed in Refs. [4,5], the following formula is obtained:

$$C^{(0)}(Q,K) \approx 1 + \frac{|\widetilde{S}(Q,K)|^2}{|\widetilde{S}(0,K)|^2}, \tag{1}$$

where $C^{(0)}(Q,K)$ is the two-particle momentum correlation function, $Q$ is the pair relative four-momentum, $K$ is the pair average four-momentum, the superscript $(0)$ denotes the neglection of final-state interactions, and $\widetilde{S}(Q,K)$ is the Fourier transform of the source with

$$\widetilde{S}(Q,K) = \int S(x,K)e^{iQx}d^4x. \tag{2}$$

Equation (1) implies that $C^{(0)}(Q=0,K) = 2$. In previous measurements, it was found, however, that $C^{(0)}(Q \to 0,K) < 2$. This result can be understood via the core–halo model [25,26], wherein the source is divided into two parts, a core of primordial hadrons and a halo of long-lived resonances. The halo is experimentally unresolvable due to its large size, which leads to small momentum in Fourier space. If $S$ represents only the core part of the source, its connection to the correlation function becomes

$$C^{(0)}(Q,K) \approx 1 + \lambda \frac{|\widetilde{S}(Q,K)|^2}{|\widetilde{S}(0,K)|^2}, \tag{3}$$

where $\lambda$ is the square of the core fraction, and it is often called the correlation strength parameter.

Using Equation (3), a theoretical formula for $C^{(0)}(Q,K)$ can be calculated by assuming a given source distribution. In this analysis, a generalization of the Gaussian distribution, the so-called spherically symmetric Lévy alpha-stable distribution [17], was assumed for the spatial part of the source. This distribution is defined by the following Fourier transform in three dimensions:

$$\mathcal{L}(\boldsymbol{r};\alpha,R) = \frac{1}{(2\pi)^3} \int \mathrm{d}^3\boldsymbol{q}\, e^{i\boldsymbol{q}\boldsymbol{r}} e^{-\frac{1}{2}|\boldsymbol{q}R|^\alpha}, \tag{4}$$

where $\boldsymbol{q}$ is an integration variable, $\boldsymbol{r}$ is the variable of the distribution, $\alpha$ and $R$ are parameters; the Lévy stability index and the Lévy scale parameter, respectively. The $\alpha$ parameter describes the shape of the distribution, with $\alpha = 2$ corresponding to the Gaussian and $\alpha = 1$ to the Cauchy case. The $R$ parameter describes the spatial scale of the source, as it is proportional to the full width at half maximum. There are many possible reasons [18–24] behind the appearance of the Lévy distribution in heavy-ion collisions, but these possibilities are still under investigation by the community. In case of a spherically symmetric Lévy source, the two-particle correlation function has the form [19]

$$C^{(0)}(q) = 1 + \lambda e^{-(qR)^\alpha}, \tag{5}$$

where $q = |\boldsymbol{Q}|$ is the magnitude of the spatial part of $Q$.

In the above formulas, the presence of final-state interactions was neglected. In the case of charged particles, the most important final-state interaction is the Coulomb interaction, which is usually taken into account in the form of a Coulomb correction $K_C(q;R,\alpha)$ [27–29]. Using the Bowler–Sinyukov method [30], one obtains

$$C(q) = 1 - \lambda + \lambda(1 + e^{-(qR)^\alpha})K_C(q;R,\alpha). \tag{6}$$

In this analysis, the $R$ and $\alpha$-dependent Coulomb correction, calculated in Ref. [31], was utilized. A formula based on Equation (6) was used for fitting to the measured correlation functions.

### 3. Measurement Details

The used data sample contains $4.27 \times 10^9$ PbPb events at a center-of-mass energy per nucleon pair of $\sqrt{s_{\mathrm{NN}}} = 5.02$ TeV, recorded by the CMS experiment in 2018. The detailed description of the CMS detector system can be found in Ref. [32]. For the analysis, only events with precisely one nucleus–nucleus collision were used, where the longitudinal distance of the interaction point from the center of the detector was also less than 15 cm. Further event selections were applied to reject events from beam–gas interactions and nonhadronic collisions [33]. The individual tracks were filtered based on their transverse momentum, pseudorapidity, distance to the vertex, the goodness of the track fit, and the number of hits in the tracking detectors.

Particle identification in central PbPb collisions is not possible with the CMS detector; therefore, all charged tracks passing the other selection criteria were used. The majority of these charged particles are pions [34], so the pion mass was assumed for all of them. The largest contamination is caused by kaons and protons [34], and this effect is discussed in Section 4.

Measuring two-particle Bose–Einstein correlation functions means measuring pair distributions. Besides the quantum-statistical effects, these pair distributions are influenced by detector acceptance, kinematics, and other phenomena. In order to remove these unwanted effects, the correlation function is calculated as the normalized ratio of two distributions, the actual (signal) distribution $A(q)$, and the background distribution $B(q)$, with

$$C(q) = \frac{A(q)}{B(q)} \frac{\int B(q)dq}{\int A(q)dq}, \tag{7}$$

where the integrals are calculated over a range where the quantum-statistical effects are not present. The $A(q)$ distribution contains all same charged pairs of a given event, while the $B(q)$ distribution contains all same charged pairs of a mixed event. This mixed event is obtained by randomly selecting particles from different events, as detailed in Refs. [11,35]. For the validity of Equation (7), it was assumed that the produced particles had a uniform rapidity distribution [36].

In the measurement of $C(q)$, the $q$ variable is taken as the magnitude of the relative momentum in the longitudinally comoving system (LCMS), where the longitudinal component of the average momentum is zero. This coordinate system was chosen because, in earlier measurements, the source was found to be approximately spherically symmetric in this frame [6]. The measurement is carried out up to $q = 8$ GeV/$c$ in 6 centrality (0–60%) and 24 average transverse momentum $K_{\mathrm{T}}$ (0.5–1.9 GeV/$c$) classes, separately for positively and negatively charged pairs. In order to remove the merging and splitting effects caused by the finite resolution of the tracking detectors, a pair selection was applied. These artifacts were limited to a region with small $\Delta\eta$ and $\Delta\phi$; therefore, each pair had to satisfy the following condition:

$$\left(\frac{|\Delta\eta|}{0.014}\right)^2 + \left(\frac{|\Delta\phi|}{0.022}\right)^2 > 1, \tag{8}$$

where $\Delta\eta$ is the pseudorapidity difference and $\Delta\phi$ is the azimuthal angle difference. Tracking efficiency correction factors were also utilized when measuring the $A(q)$ and $B(q)$ distributions.

Even after removing most of the non-quantum-statistical effects by taking the ratio of $A(q)$ and $B(q)$, a structure was observed in $C(q)$ at large $q$ values, where the quantum-statistical effects were not present. This long-range background can be the result of phenomena such as energy and momentum conservation, resonance decays, bulk flow [15], and minijets [15]. To remove any potential influence of the long-range background on the



low $q$ region where the Bose–Einstein peak is present, $C(q)$ was divided by a background function $BG(q)$, resulting in the double-ratio correlation function $DR(q)$:

$$DR(q) = \frac{C(q)}{BG(q)}. \tag{9}$$

The explicit form of $BG(q)$ was determined by fitting the following empirically determined formula [15,37,38] to the large $q$ part of $C(q)$:

$$BG(q) = N\left(1 + \alpha_1 e^{-(qR_1)^2}\right)\left(1 - \alpha_2 e^{-(qR_2)^2}\right), \tag{10}$$

where $N, \alpha_1, \alpha_2, R_1, R_2$ are fit parameters with no physical meaning.

The $DR(q)$ distributions were fitted with the following formula based on Equation (6):

$$DR(q) = N(1 + \epsilon q)\left[1 - \lambda + \lambda(1 + e^{-(qR)^\alpha})K_C(q; R, \alpha)\right], \tag{11}$$

where $N$ is a normalization parameter and a possible residual linear background is allowed through the $\epsilon$ parameter. The fits were performed using the MINUIT2 package [39,40] and the statistical uncertainties were calculated with the MINOS algorithm [39,40]. The lower and upper fit limits were determined individually in each centrality and $K_T$ class by selecting the limits resulting in the best fit. The goodness of fit was measured by the confidence level, calculated from the $\chi^2$ and the number of degrees of freedom of the fit. This confidence level was in the statistically acceptable range ($>0.1\%$) for each fit. An example fit is shown in Figure 1. In the region below approximately $q = 0.05\,\text{GeV}/c$, the measured data are not reliable due to the finite momentum resolution and pair reconstruction efficiency of the detectors; consequently, that region was not used for fitting.

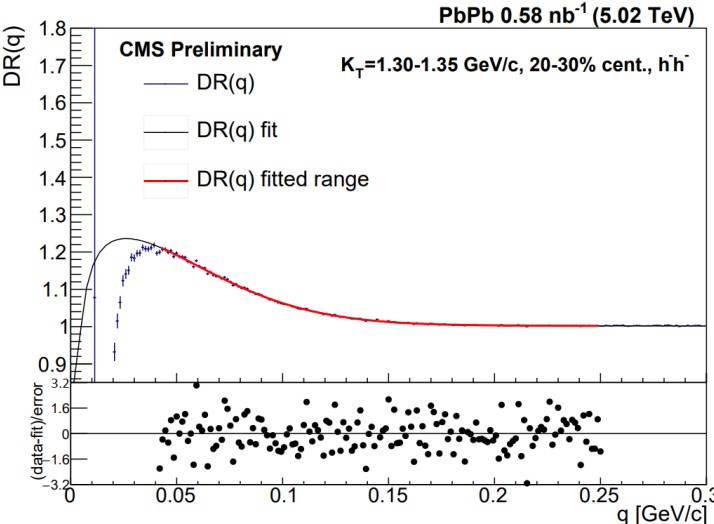

**Figure 1.** An example fit to the double-ratio correlation function $DR(q)$ of negatively charged hadrons [41]. The fitted function is shown in black, while the red overlay indicates the range used for the fit. The $K_T$ and centrality class is shown in the legend. The lower panel indicates the deviation of the fit from the data.

The systematic uncertainties of $R, \alpha$, and $\lambda$ were determined by individually changing each of the analysis settings to slightly larger and smaller values, and conducting the whole analysis procedure again. The deviations from the nominal results were then added in quadrature, resulting in the full systematic uncertainty. The considered analysis settings were the centrality calibration, the vertex selection, the different track selection criteria, the pair selection, and the fit limits. Out of these, the dominant sources of systematic uncertainty were the fit limits. The full systematic uncertainty was separated into correlated

and uncorrelated parts, so that the latter could be taken into account when fitting to the parameters.

## 4. Results and Discussion

As mentioned before, the parameters $\alpha$, $R$, and $\lambda$ were measured separately for positively and negatively charged hadron pairs. As not much difference was observed between the two cases, some of the results for negatively charged pairs are shown only in Appendix A.

The measurement was carried out in $K_T$ classes, but in order to facilitate the comparison with previous measurements and with theory, the parameters are presented as functions of the transverse mass $m_T$, defined as

$$m_T = \sqrt{\frac{K_T^2}{c^2} + m^2},\tag{12}$$

where $m$ is the mass of the investigated particle species. Although all charged tracks were used in the analysis, the pion mass was used for $m$, since above 90% of the identical particle pairs were pion pairs.

The measured $\alpha$ values are shown in Figure 2 as a function of $m_T$, for positively charged pairs. Within uncertainties, most of the values are between 1.6 and 2.0, meaning that the source follows the general Lévy distribution, instead of the Gaussian. However, the deviation from the Gaussian case is not as large as it was found for 0–30% centrality AuAu collisions at $\sqrt{s_{NN}} = 200$ GeV [11], where a mean value for $\alpha$ of 1.207 was obtained for pion pairs with $|\eta| < 0.35$ and $228 < m_T < 871$ MeV/$c^2$. For a given centrality class, $\alpha$ is almost constant with $m_T$. The average of $\alpha$ ($\langle\alpha\rangle$) is indicated in Figure 2 for each centrality class, and it is shown in Figure 3 as a function of the average number of participating nucleons in the collision ($\langle N_{part}\rangle$), for both positively and negatively charged pairs. The $\langle N_{part}\rangle$ values were calculated for each centrality class [42], with a larger value corresponding to a more central case. The $\langle\alpha\rangle$ values show a monotonic increasing trend with $\langle N_{part}\rangle$, which means that the shape of the source is $\langle N_{part}\rangle$ (or equivalently, centrality) -dependent. The shape is closer to the Gaussian distribution in case of more central events. The $\langle\alpha\rangle$ values are slightly higher for positively charged pairs, although the deviations are within systematic uncertainties.

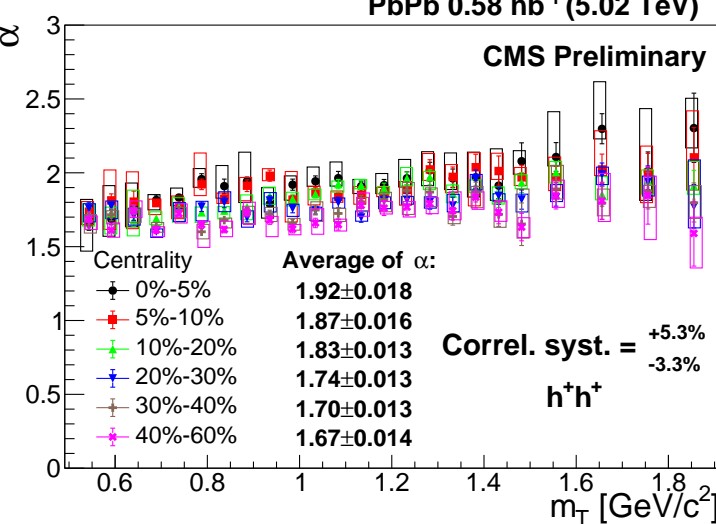

**Figure 2.** The Lévy stability index $\alpha$ versus the transverse mass $m_T$ in different centrality classes for positively charged hadron pairs [41]. The error bars are the statistical uncertainties, while the boxes indicate the uncorrelated systematic uncertainties. The correlated systematic uncertainty is shown in the legend.

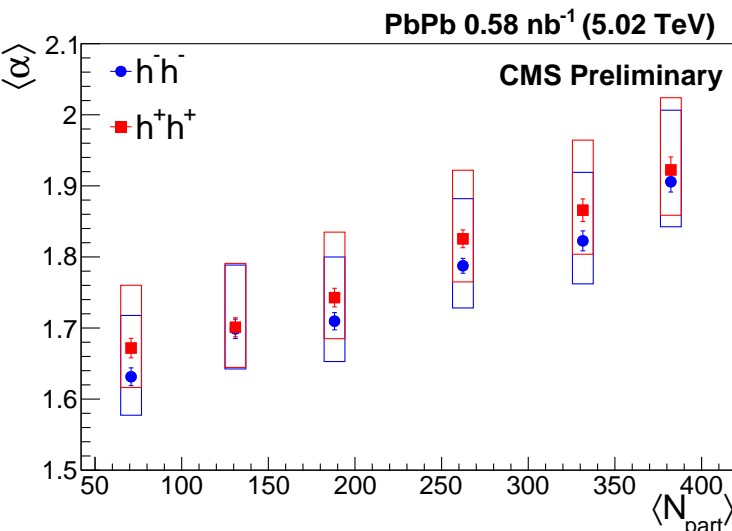

**Figure 3.** The average Lévy stability index $\langle\alpha\rangle$ versus $\langle N_{\text{part}}\rangle$ in different centrality classes for positively and negatively charged hadron pairs [41]. The error bars are the statistical uncertainties, while the boxes indicate the systematic uncertainties.

The measured $R$ values are shown in Figure 4 as a function of $m_T$ for positively charged pairs. A decreasing trend with $m_T$ and as the collisions become more peripheral is observed, with the values ranging between 1.6 and 5.8 fm. The centrality dependence confirms the geometrical interpretation of the $R$ parameter, because a smaller source size is expected in case of more peripheral collisions. To further investigate the $m_T$ dependence of $R$, $1/R^2$ was plotted as a function of $m_T$, as shown in Figure 5. In case of a Gaussian source, hydrodynamic models [7,43] predict the linear scaling

$$\frac{1}{R^2} = Am_T + B, \tag{13}$$

where $A$ and $B$ are parameters with physical meaning. The slope $A$ is connected to the Hubble constant ($H$) of the QGP with [7,44]

$$A = \frac{H^2}{T_f}, \tag{14}$$

where $T_f$ is the freeze-out temperature. The intercept $B$ is connected to the size of the source ($R_f$) at freeze-out with [7,44]

$$B = \frac{1}{R_f^2}. \tag{15}$$

In order to verify whether the linear scaling also holds in the Lévy case, a linear fit was performed for each centrality class using Equation (13). The statistical uncertainty and the uncorrelated systematic uncertainty of $1/R^2$ was added in quadrature and used for determining the $\chi^2$ of the fits. In this way, the confidence levels were statistically acceptable for each centrality class, showing that a hydrodynamic-like scaling holds for a Lévy source as well. The fitted lines are shown in Figure 5, and the fit parameters ($A$ and $B$) are shown in Figure 6 as functions of $\langle N_{\text{part}}\rangle$, for both positively and negatively charged pairs. By assuming a constant freeze-out temperature of $T_f = 156$ MeV [45], the Hubble constant falls between 0.12 $c/$fm and 0.18 $c/$fm. Due to the fact that the $A$ parameter decreases toward more central collisions (larger $\langle N_{\text{part}}\rangle$), the Hubble constant also decreases, making the speed of the expansion lower in central collisions. The $B$ parameter has a negative value in each case, which makes it impossible to calculate a freeze-out size using Equation (15). The reasons behind a negative intercept and the interpretation of this result are currently

unknown. This may be connected to fluctuations in the initial state [46] which were not taken into account in the hydrodynamic models.

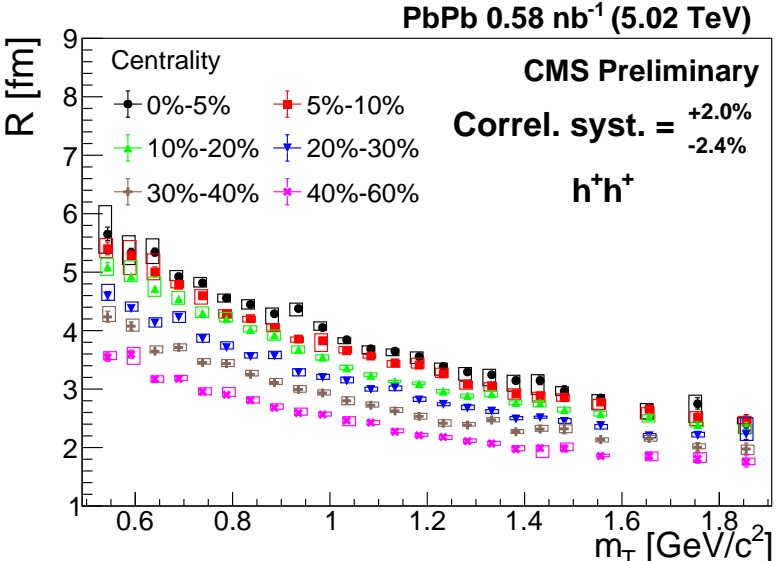

**Figure 4.** The Lévy scale parameter $R$ versus $m_T$ in different centrality classes for positively charged hadron pairs [41]. The error bars are the statistical uncertainties, while the boxes indicate the uncorrelated systematic uncertainties. The correlated systematic uncertainty is shown in the legend.

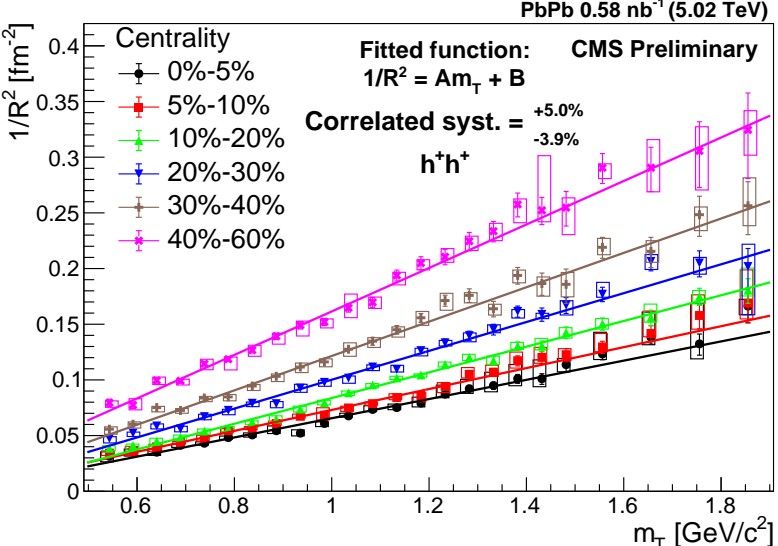

**Figure 5.** The inverse square of the Lévy scale parameter $R$ versus $m_T$ in different centrality classes for positively charged hadron pairs [41]. The error bars are the statistical uncertainties, while the boxes indicate the uncorrelated systematic uncertainties. The correlated systematic uncertainty is shown in the legend. A line is fitted to the data for each centrality.

The measured $\lambda$ values are shown in the upper panel of Figure 7 as a function of $m_T$, for positively charged pairs. A decreasing trend with $m_T$ as the collisions became more central is observed. In case of identified particles, $\lambda$ is the square of the ratio of core particles. Due to the lack of particle identification, our sample contained particles other than pions, mostly kaons and protons. As a result of this contamination, $\lambda$ was suppressed by a factor of the square of the pion fraction. The pion fraction was measured by the ALICE Collaboration [34], and it decreased with $m_T$, resulting in the decreasing trend of $\lambda$ in the upper panel of Figure 7. For the $\alpha$ and the $R$ parameters, a characteristic $m_T$ dependence

was observed; thus, these parameters could not have been influenced by the $m_T$-dependent effect of the lack of particle identification. To remove the effect of the contamination from $\lambda$, the $\lambda^*$ parameter was introduced by rescaling $\lambda$ with the square of the pion fraction:

$$\lambda^* = \frac{\lambda}{(N_{pion}/N_{hadron})^2}. \tag{16}$$

The rescaled correlation strength $\lambda^*$ is shown in the lower panel of Figure 7. Compared to $\lambda$, the decreasing trend with $m_T$ is no longer shown in the data, suggesting that it was caused purely by the lack of particle identification. The centrality dependence, on the other hand, remained the same, which means that the fraction of core pions is smaller in more central collisions.

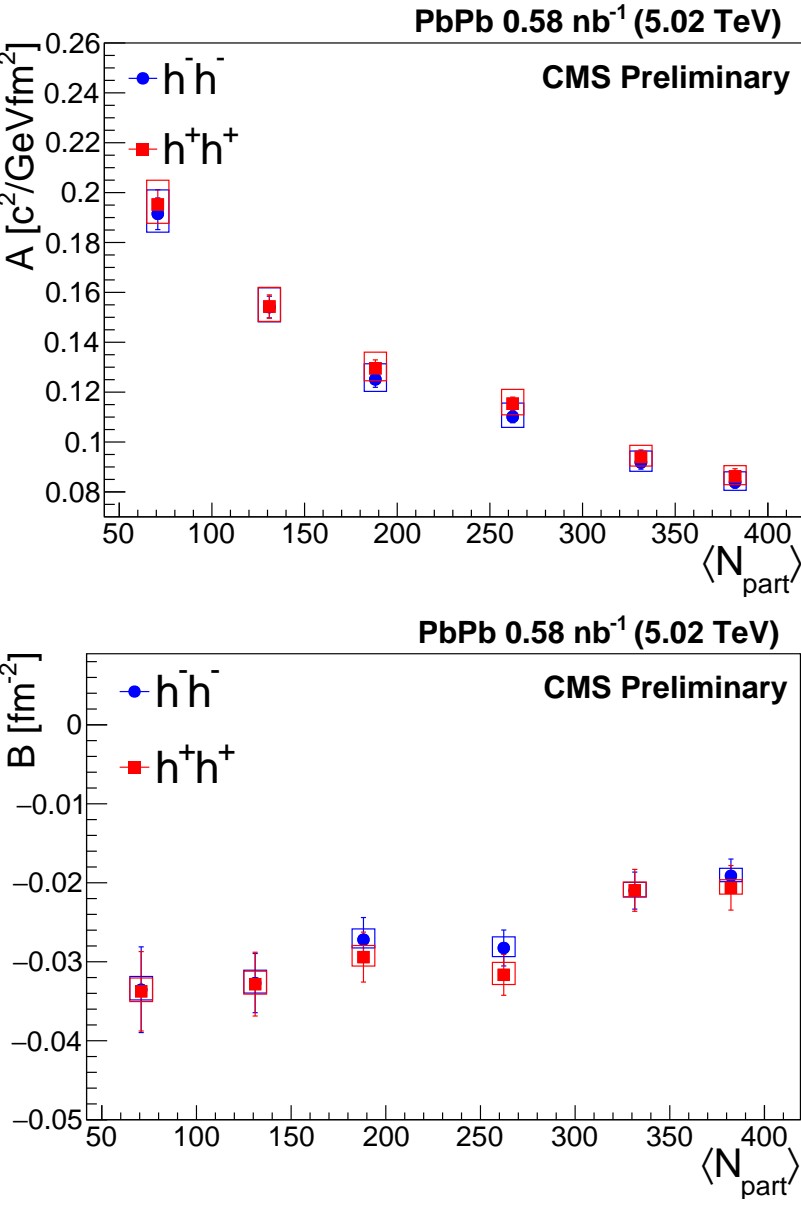

**Figure 6.** The two fit parameters from the linear fit: the slope $A$ (**upper**) and the intercept $B$ (**lower**) versus $\langle N_{part} \rangle$ for negatively and positively charged hadron pairs [41]. The error bars are the statistical uncertainties, while the boxes indicate the systematic uncertainties.

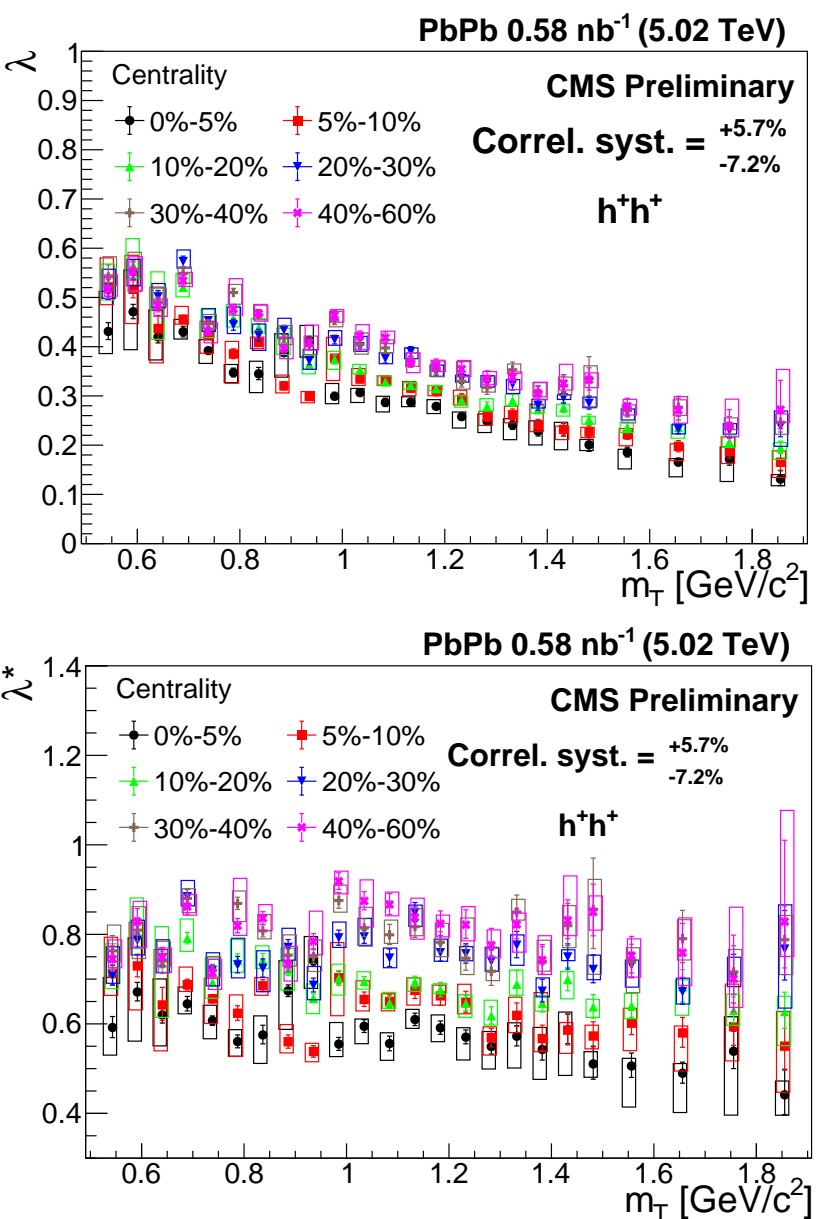

**Figure 7.** The correlation strength $\lambda$ and the rescaled correlation strength $\lambda^*$ versus $m_T$ in different centrality classes for positively charged hadron pairs [41]. The error bars are the statistical uncertainties, while the boxes indicate the uncorrelated systematic uncertainties. The correlated systematic uncertainty is shown in the legend.

## 5. Conclusions

In this paper, a centrality-dependent Lévy HBT analysis of two-particle Bose–Einstein correlations was presented, using $\sqrt{s_{NN}} = 5.02$ TeV PbPb collision data recorded by the CMS experiment. The measured correlation functions were described by the assumption of a Lévy alpha-stable source distribution. Three source parameters, the Lévy stability index $\alpha$, the Lévy scale parameter $R$, and the correlation strength $\lambda$ were determined, and their centrality and transverse mass ($m_T$) dependence was investigated.

The $\alpha$ parameter was found to be centrality-dependent, but constant in $m_T$, with the average values ranging between 1.6 and 2.0. A decreasing trend with $m_T$ and as the collisions become more peripheral was observed for the $R$ parameter, which could be explained by the hydrodynamic-like scaling and the geometrical interpretation, respectively. The $\lambda$ parameter showed a decreasing trend with $m_T$, but after removing the effects of the

lack of particle identification, a constant behavior was obtained. A decrease toward more central collisions was also observed for $\lambda$.

**Funding:** B. Kórodi was supported by the ÚNKP-21-2 New National Excellence Program of the Ministry for Innovation and Technology from the source of the National Research, Development and Innovation Fund. This research was supported by the NKFIH OTKA K-138136 and K-128713 grants.

**Data Availability Statement:** The data presented in this study are available on request from the corresponding author. The data are not publicly available.

**Conflicts of Interest:** The author declares no conflict of interest.

## Abbreviations

The following abbreviations are used in this manuscript:

| | |
|---|---|
| QGP | quark–gluon plasma |
| LHC | Large Hadron Collider |
| HBT | Hanbury Brown and Twiss |
| PbPb | lead–lead |
| CMS | Compact Muon Solenoid |
| AuAu | gold–gold |

## Appendix A. Results for Negatively Charged Pairs

The results for negatively charged hadron pairs are presented. Due to the fact that they are very similar to the results for positively charged pairs presented in Section 4, the interpretations of these results are the same. The Lévy stability index $\alpha$ is shown as a function of $m_T$ in Figure A1. The Lévy scale parameter $R$ and its inverse square $1/R^2$ are shown as functions of $m_T$ in Figures A2 and A3, respectively. The correlation strength $\lambda$ and the rescaled correlation strength $\lambda^*$ are shown as functions of $m_T$ in Figure A4.

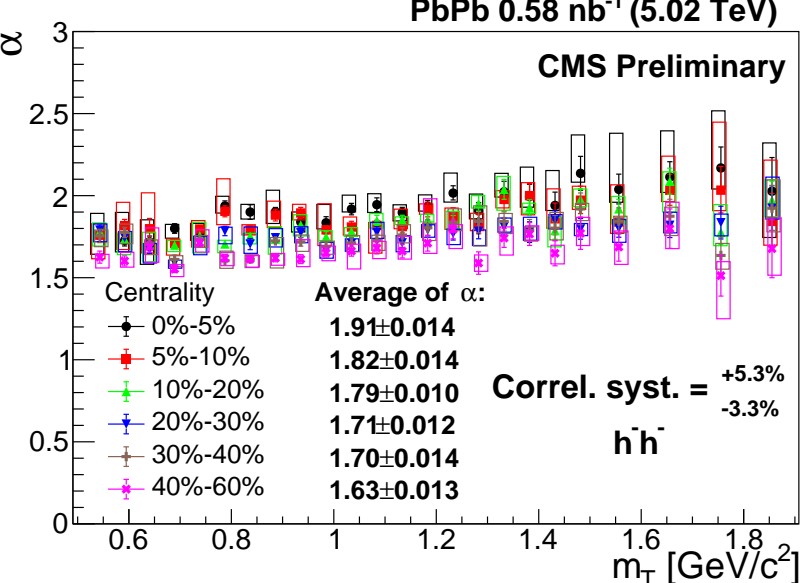

**Figure A1.** The Lévy stability index $\alpha$ versus the transverse mass $m_T$ in different centrality classes for negatively charged hadron pairs [41]. The error bars are the statistical uncertainties, while the boxes indicate the uncorrelated systematic uncertainties. The correlated systematic uncertainty is shown in the legend.

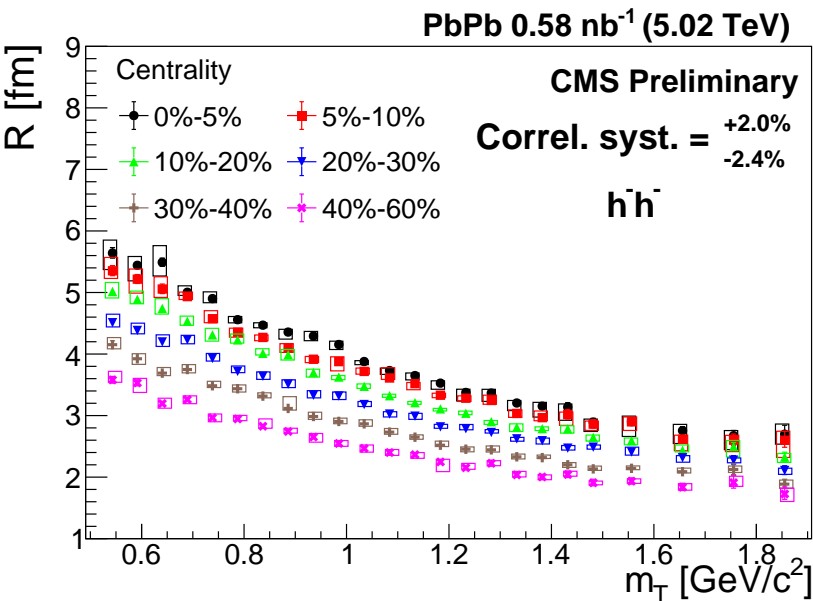

**Figure A2.** The Lévy scale parameter $R$ versus $m_T$ in different centrality classes for negatively charged hadron pairs [41]. The error bars are the statistical uncertainties, while the boxes indicate the uncorrelated systematic uncertainties. The correlated systematic uncertainty is shown in the legend.

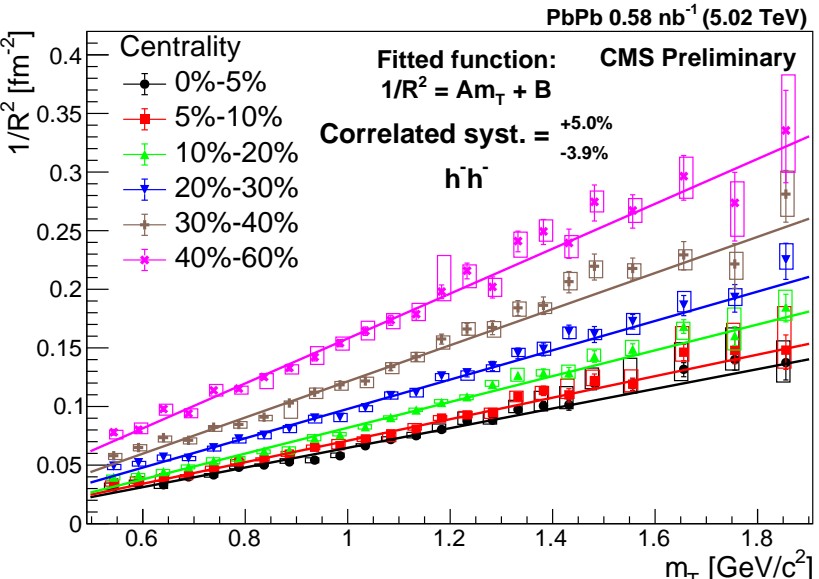

**Figure A3.** The inverse square of the Lévy scale parameter $R$ versus $m_T$ in different centrality classes for negatively charged hadron pairs [41]. The error bars are the statistical uncertainties, while the boxes indicate the uncorrelated systematic uncertainties. The correlated systematic uncertainty is shown in the legend. A line is fitted to the data for each centrality.

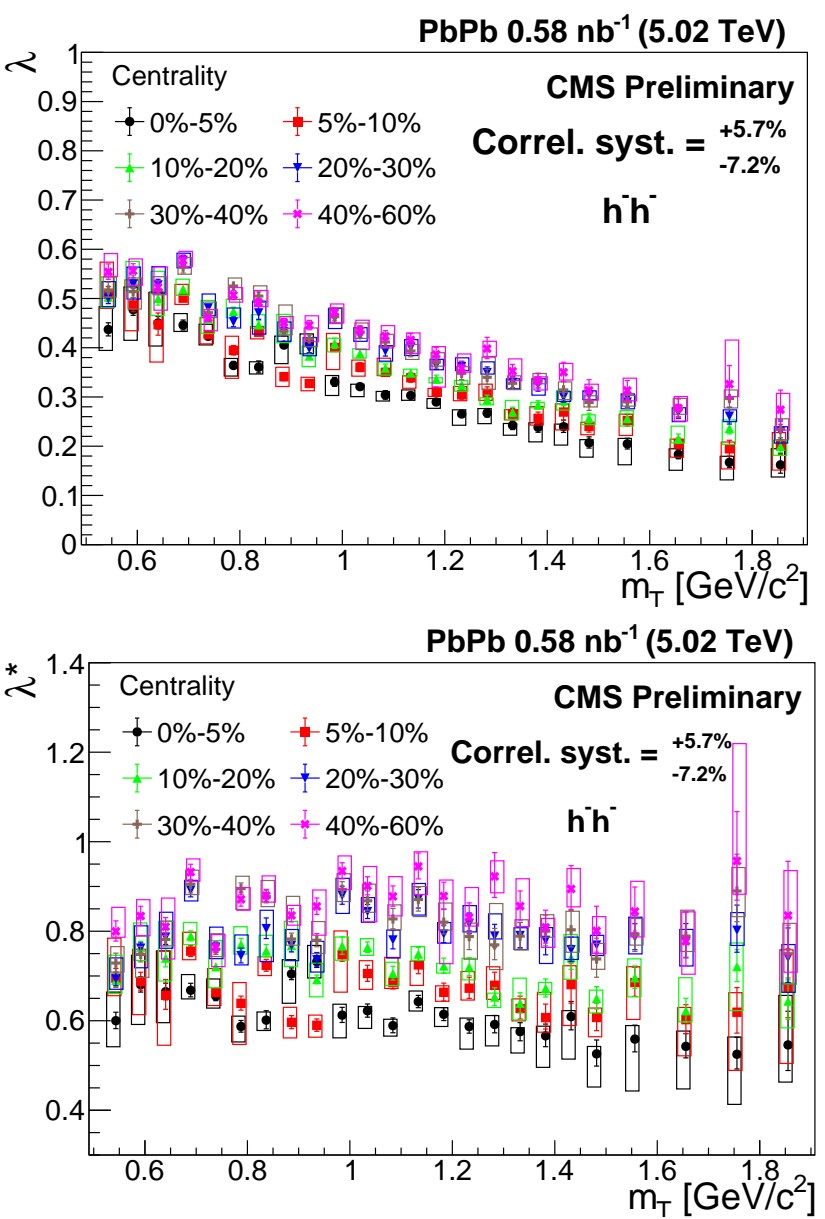

**Figure A4.** The correlation strength $\lambda$ and the rescaled correlation strength $\lambda^*$ versus $m_T$ in different centrality classes for negatively charged hadron pairs [41]. The error bars are the statistical uncertainties, while the boxes indicate the uncorrelated systematic uncertainties. The correlated systematic uncertainty is shown in the legend.

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
