# Peer review of "Centrality-Dependent Lévy HBT Analysis in sNN=5.02 TeV PbPb Collisions at CMS"

_universe, doi:10.3390/universe9070318_

Round 1

Reviewer 1 Report

The author presents an HBT analysis of Bose-Einstein two-particle momentum correlation functions with Lévy sources, based on CMS data at $sqrt{s}=5.02$ TeV. The analysis is structured to obtain results as functions of transverse mass and centrality. Making use of the expected form of the correlation functions with Lévy sources known from existing literature, the author performs fits that determine the shape and size of the emitting source, as well as the relative fraction of the emitting core in a core-corona picture.

The main result is that the source shape is found to be neither Gaussian, nor Cauchy, exception made for high-multiplicity events. Moreover, the extracted source size shows a dependence on the transverse mass that is compatible with hydrodynamic behavior.

Finally, the correlation strength, which also indicates the core fraction in the source, is found to be independent of the transverse mass after suitable corrections, while showing a decreasing trend towards higher centrality.

I find that the work is well presented, and the analysis is carried out rigorously. The introduction is clear and sufficiently exhaustive. I recommend this work for publication, after the following two points are addressed:

a.) In the introduction, the author states that one of the motivations for this work is to study the source shape at higher energies than previously considered, and compare to results at lower collision energies. However, this comparison is not discussed in the text. I think this point deserves a mention in the discussion of the results.

b.) From the analysis of the core fraction, it emerges that in central collisions the fraction of core pions is smaller than in peripheral ones. This seems counterintuitive, is there a simple interpretation for this?

Reviewer 2 Report

The manuscript is very well written and the result is very interesting. I have just one request to be included. In the lines 189-201, the effect of no-PID is discussed in the context of the lambda value measurement. I would like to know if there is a sizable effect on the R_inv by no-PID, and if it is, I think that should be included as a part of systematic uncertainties. Please add some discussion on this. Other that this, the manuscript is fine for publication.

Reviewer 3 Report

The article is devoted to an actual topic of great physical interest - the femtoscopic studies of the space-time geometry of heavy ion collisions at high energies. The paper makes a good impression, it is well written, contains a fairly detailed introduction to the subject with the necessary references, and is well formatted. It presents important new physical results, the most interesting of which is the investigation of the Lévy scale parameter R as a function of the average transverse momentum of the pair  (per particle) for various collision centrality classes, which is presented as a function of the transverse mass, , to facilitate comparison with previous measurements and with theory.

            It is important that the centrality dependence observed in Fig. 4 confirms the geometric interpretation of the parameter R as the spatial size of the source decreasing for more peripheral collisions. In Fig. 5 we see that the linear form of the dependence of  on  confirms the predictions of hydrodynamic models for each centrality class. This allows the authors to extract the so-called Hubble constant H of the QGP, which characterizes the QGP expansion rate, assuming a constant freeze-out temperature of 156 MeV. Under this assumption, the Hubble constant occurs decreasing towards more central collisions, what corresponds to a lower expansion speed in central collisions than in peripheral ones.

            Unfortunately, when the authors try to extract the size of the source at freeze-out, , using relation (15), , they fail to do so because of the negative values of B. The reason for this, in our opinion, is that formula (13) was obtained within the framework of purely hydrodynamic models [7, 39], which do not take into account the fluctuation of the initial conditions, which is inevitable in real experimental data. An example is the so-called "volume" fluctuations - fluctuations in the number of participating nucleons, due to event-by-event fluctuations of the impact parameter. These fluctuations take place even within events belonging to the same centrality class. They slightly affect the mean values of the observables, but make a significant contribution to their variances and correlation functions, which are considered in the paper.

              It is clear that this contribution from the event-by-event fluctuation of the initial conditions to the correlation function must be taken into account in addition to purely hydrodynamic correlations. It is possible that taking into account this additional contribution will lead to positive values of B. But this is rather a wish for further work for theorists. It is important that this experimental work explicitly reveals this problem. Experimentally, this issue can be investigated by narrowing the width of the centrality classes, which, in principle, should reduce the contribution of fluctuations of the initial conditions to the correlation function, although this contribution cannot be completely eliminated [Sputowska, I. Proceedings 2019, 10(1), 14].

            One more remark concerns the use of formula (7) for the experimental determination of the value of the correlation function. A similar formula, where the denominator uses the procedure for mixing tracks from different events, is widely used in experimental work, since it allows removing unwanted experimental effects as detector acceptance and efficiency, kinematics, and so on. However, a more general and profound definition of a two-particle correlation function is the ratio of a two-particle distribution to the product of two one-particle distributions. This definition can be reduced to formula (7) only if certain conditions are met, in particular, it is necessary to have the uniform distribution of the produced particles in rapidity (see e.g. formulas (16), (21), (24) and (29) in [Vechernin, V., Andronov, E. Universe 2019, 5, 15]).

            But even in this case, formula (7) cannot reproduce the absolute value of a strictly defined two-particle correlation function, since, due to uncertainties in the event mixing procedure, it cannot correctly take into account the contribution to the two-particle correlation function arising from the mentioned volume fluctuations. (See for details Appendix C in [Vechernin, V. Forward–backward correlations between multiplicities in windows separated in azimuth and rapidity. Nucl. Phys. A 2015, 939, 21], the article also demonstrates that we can extract the absolute value of the two-particle correlation function, taking into account initial condition fluctuation, by measuring the magnitude of the so-called forward-backward correlations in two narrow observation windows separated in rapidity and azimuth.)

            In this regard, there remains an uncertainty that the procedure of dividing by the background function and passing to the two-fold correlation function used in the article makes it possible to save only the quantum-statistical contribution and completely exclude the contribution to the correlation function arising from fluctuations of the initial conditions. Since if there are two causes of correlations, the resulting correlation function will not be equal to the product of the correlation function arising from each of them. In this case, the purely hydrodynamic formula (13) will be incomplete for the analysis of the experimental correlation function containing the residual contribution from fluctuations of the initial conditions, leading to the negative value of B.

            Some disadvantage of work in which particle identification (PID) is not carried out is the use of a transverse mass. In this case, the authors have to assign pion masses to all particles. At the end of the article, the authors try to effectively take into account the influence of this approximation on the results obtained. Figure 7 shows that it is rather noticeable. So, after taking it into account, the parameter λ no longer depends on the transverse mass. Most likely, that a more accurate analysis taking into account PID, will also affect other parameters, including B. Therefore, it would be very interesting to continue these studies already with PID.

              There is also one technical note.

In Fig.5, the statistical and systematic errors for each point are approximately equal to each other. But in the lower panel of Fig. 6, the statistical errors turn out to be larger than the systematic ones. It's right? They seem to get by continuing the straight lines in Fig. 5 to the point  according to the same procedure. (Moreover, in the upper panel of Fig. 6, statistical and systematic are still approximately equal).

            In general, the work seems to be very interesting. It contains important new results that will undoubtedly serve as an impetus for new experimental and theoretical studies. I think that they can be published in the journal after very minor corrections.

(Please see the attached .pdf file for the correct display of formulas.)

Round 2

Reviewer 3 Report

 Accept in present form.